# Do We Still Need Aspirin in Coronary Artery Disease?

**DOI:** 10.3390/jcm12247534

**Published:** 2023-12-06

**Authors:** Muhammad Haisum Maqsood, Glenn N. Levine, Neal D. Kleiman, David Hasdai, Barry F. Uretsky, Yochai Birnbaum

**Affiliations:** 1Department of Cardiology, DeBakey Heart and Vascular Center, Methodist Hospital, Houston, TX 77030, USA; haisumbajwa@live.com; 2The Section of Cardiology, Baylor College of Medicine, Houston, TX 77030, USA; glevine@bcm.edu; 3Department of Cardiology, Section of Interventional Cardiology, Houston Methodist DeBakey Heart Center, Houston, TX 77030, USA; nkleiman@houstonmethodist.org; 4Department of Cardiology, Rabin Medical Center, Tel Aviv University, Petah Tikva 49200, Israel; dhasdai@post.tau.ac.il; 5Central Arkansas Veterans Health System, University of Arkansas for Medical Sciences, Little Rock, AR 72205, USA; buretsky@gmail.com

**Keywords:** acute coronary syndrome, aspirin, clopidogrel, P2Y12 inhibitors, percutaneous coronary intervention, primary prevention, secondary prevention, stable coronary artery disease, ticagrelor

## Abstract

Aspirin has for some time been used as a first-line treatment for acute coronary syndromes, including ST-elevation myocardial infarction, for secondary prevention of established coronary disease, and for primary prevention in patients at risk of coronary artery disease. Although aspirin has been in use for decades, the available evidence for its efficacy largely predates the introduction of other drugs, such as statins and P2Y12 inhibitors. Based on recent trials, the recommendation for aspirin use as primary prevention has been downgraded. In addition, P2Y12 inhibitors given as a single antiplatelet therapy have been associated with a lower incidence of bleeding than dual antiplatelet therapy in combination with aspirin in patients with stable and unstable coronary artery disease. The aim of this review is to discuss the role of aspirin considering the available evidence for primary prevention, secondary prevention for stable coronary artery disease or acute coronary syndromes, and after percutaneous coronary intervention or coronary artery bypass revascularization.

## 1. Introduction

Aspirin has for some time been a first-line treatment for acute coronary syndromes (ACS), including ST-elevation myocardial infarction (STEMI), a primary and secondary prevention in patients at risk for or established coronary artery disease (CAD), and for after percutaneous coronary intervention (PCI) and coronary artery bypass grafting surgery (CABG) [1,2,3,4,5]. Aspirin has been in use for decades, and the available evidence of its efficacy is mainly based on studies conducted before the introduction of other drugs that reduce morbidity and/or mortality, such as HMG-CoA reductase inhibitors (statins) and P2Y12 inhibitors [6]. Recently, aspirin use for primary prevention has been downgraded by the American College of Cardiology/American Heart Association and U.S. Preventive Services Task Force [3,5], based on data from trials including Aspirin to Reduce Risk of Initial Vascular Events (ARRIVE) [7], A Study of Cardiovascular Events in Diabetes (ASCEND) [8], and Aspirin in Reducing Events in the Elderly (ASPREE) [9]. In addition, several recent studies have suggested that P2Y12 inhibitors (clopidogrel or ticagrelor) as antiplatelet monotherapy have better efficacy than aspirin in patients with stable CAD or in patients after an ACS event [10,11,12,13,14,15,16,17,18]. Further, European Society of Cardiology guidelines for acute coronary syndrome have been published recently, reflecting these new findings [19]. The aim of this review is to evaluate the current role of aspirin in light of the available evidence for primary prevention, loading, and maintenance dosing after ACS (STEMI and non-ST-elevation myocardial infarction (NSTEMI)), for coronary revascularization, and for secondary prevention.

**What is New:** For many years aspirin was recommended for primary prevention in individuals at risk for and for patients with established coronary artery disease. Those recommendations were based on studies conducted before the introduction of other therapies that significantly affect prognosis, including statins, other antiplatelet agents (mainly P2Y12 inhibitors), and coronary interventions.

Recent clinical trials have shown that aspirin does not show clinical benefits when given for primary prevention. The duration of dual antiplatelet agent therapy after ACS and/or PCI is shortened by the current guidelines, according to the results of recent trials. It also seems that in patients with stable coronary artery disease or those after PCI, long-term monotherapy with P2Y12 inhibitors is better and safer than aspirin monotherapy. The most recent practice guidelines have changed the recommendations concerning aspirin.

Yet, aspirin loading for patients with ACS (including STEMI and NSTE-ACS) is still considered a “gold-standard”. Data from animal models suggest that aspirin loading can block myocardial protection by various agents. However, as discussed in this manuscript, the original evidence supporting aspirin loading is weak and has never been properly tested. It might be that aspirin loading can be skipped. Further clinical studies are needed to assess clinical outcomes in these patients without and with aspirin loading.

### 1.1. Role of Aspirin for Primary Prevention

Aspirin was used for the primary prevention of adverse cardiovascular events. However, three recent placebo-controlled trials of aspirin 100 mg–ARRIVE [7], ASPREE [9], and ASCEND [8] have failed to show the superiority of aspirin over a placebo in reducing the rate of composite ischemic outcomes, while they have shown increased bleeding risk (Table 1).

The ARRIVE trial included 12,546 patients (mean age 64 years, 70% male, all without diabetes) [7]. Patients assigned to daily aspirin had no significant difference in the primary endpoint, a composite outcome of time to first occurrence of cardiovascular death, myocardial infarction (MI), unstable angina, stroke, or transient ischemic attack (TIA) (hazard ratio, 0.96; 95% CI 0.81–1.13) compared to a placebo at a median follow up of 60 months. The ASPREE trial included 19,114 patients (mean age 74 years, 44% males, 11% diabetic patients) [9]. ASPREE also failed to show a significant difference in the composite primary endpoint of death, dementia, or persistent physical disability (hazard ratio, 0.95; 95% CI 0.83–1.08) with 100 mg aspirin daily compared to a placebo (median follow up 4.7 years). There was also no difference in MI incidence (hazard ratio, 0.93; 95% CI, 0.76–1.15) or stroke (hazard ratio, 0.95; 95% CI, 0.83 to 1.08). The ASCEND trial included 15,480 diabetic patients (mean age 63 years, 63% men) [8]. Unlike the previous two trials, this trial observed a lower risk of the primary efficacy endpoint of the first serious vascular event (MI, stroke or TIA, or death from any vascular cause, excluding any confirmed intracranial hemorrhage) (rate ratio, 0.88; 95% CI, 0.79–0.97). However, the risk of a major bleeding event (i.e., intracranial hemorrhage, sight-threatening ocular bleeding event, gastrointestinal bleeding, or other serious bleeding) was higher with aspirin (rate ratio, 1.29; 95% CI, 1.09–1.52) with a mean follow up of 7.4 years. These risks of bleeding exponentiate with advanced age and frailty [8,30].

Similarly, a recent meta-analysis of 16 trials evaluating the use of aspirin with and without statins in patients without known atherosclerotic cardiovascular disease also concluded that the bleeding risk with aspirin outweighed the prevention for MI, most likely because the basal risk of MI was reduced by statin therapy [31]. In this meta-analysis statin use reduced the protective effect of aspirin and the bleeding risk with aspirin outweighed its protective benefit. The conclusion is supported by the observation made in the study that aspirin’s ability to prevent MI decreased as the level of LDL reduction increased [31].

Based on the aforementioned trials, the 2019 ACC/AHA guidelines for primary prevention of cardiovascular disease downgraded the recommendations for aspirin use for primary prevention. The updated recommendation is that low-dose aspirin 75–100 mg may be reasonable to use in patients between ages 40 and 70 years with higher risk of atherosclerotic cardiovascular disease who are not at increased risk of bleeding, a class of recommendation (COR) IIb and level of evidence (LOE) A recommendation (Table 2) [3]. Other aspirin-related guidelines are reported in Table 2 [1,2,4]. They recommend against the use of aspirin in patients with a higher risk of bleeding or in those above the age of 70 years (COR III: harm, LOE B-R). In 2022, the US Preventive Services Task Force recommended the use of aspirin for primary prevention in patients between 40 and 59 years who have a 10% or greater risk of 10-year cardiovascular event rate (determined using the 2013 AHA risk calculator) and who are not at increased risk of bleeding [5]. It also recommended against the routine use of aspirin in patients aged greater than 60 years.

### 1.2. Loading for NSTEMI, STEMI, or Elective PCI

The AHA/ACC 2014 guidelines for the management of patients with non–ST-elevation ACS recommended that patients with NSTEMI should be given aspirin (162 to 325 mg) immediately after presentation (COR 1, LOE A) [2]. Aspirin loading, in NSTEMI-ACS or PCI in NSTEMI has not been extensively tested, apart from a trial by Theroux [34]. Enrollment included 479 patients soon after hospital admission (7.9 h after last chest pain episode) divided into early administration of aspirin (650 mg administered orally followed by 325 mg twice daily), heparin (1000 units/hour), combined aspirin and heparin, or placebo in patients with unstable angina. The incidence of MI (defined as typical chest pain unrelieved by nitroglycerin and last at least 30 min with new ST-T segment changes or appearance of new Q waves, and doubling of baseline level of creatinine kinase with high creatinine kinase MB levels) after 6 + 3 days was lower with aspirin vs. placebo (3.3% vs. 11.9%; *p* = 0.012) and combined aspirin and heparin vs. placebo (0.8% vs. 11.9%; *p* < 0.001) [34]. This trial compared combined aspirin loading and maintenance to a placebo and showed a substantial benefit in preventing MI. However, the effects of aspirin loading were not tested. Thus, we do not know if the beneficial effects of aspirin can be attributed to the loading dose, the maintenance dose, or both.

According to the 2017 European Society of Cardiology and 2013 ACC/AHA guidelines for the management of acute MI in patients presenting with ST-segment elevation, aspirin loading (162–325 mg) is a COR 1 LOE B indication [1,35]. Aspirin loading is derived from the landmark Second International Study of Infarct Survival (ISIS-2) which randomized 17,187 patients with suspected acute MI within 24 h after the onset to four arms: aspirin (first tablet sucked or chewed followed by an oral dose 160 mg/day for one month), 1 h intravenous infusion of 1.5 million U of streptokinase, both aforementioned treatments, or neither [36]. In this trial, aspirin significantly reduced vascular mortality, non-fatal reinfarction, or non-fatal stroke, without increasing the risk of cerebral hemorrhage and bleeding requiring transfusion compared to placebo through a 5-week follow up period. However, ISIS-2 did not address the question of whether aspirin should be given as a loading dose, as there was no group treated with a maintenance dose of aspirin without acute loading.

The 2021 ACC/AHA/Society for Cardiovascular Angiography and Interventions’ guidelines for coronary artery revascularization recommended aspirin loading in patients before PCI, which is a COR I and LOE B-R [37]. These guidelines were based mainly on trials conducted in the 1980s. Barnathan et al. conducted a retrospective trial on 300 consecutive patients without angiographic evidence of thrombus who underwent elective PCI between 1980 and 1985 and classified patients into three groups: no aspirin or dipyridamole (n = 121), standard treatment (aspirin with or without dipyridamole but did not receive both drugs before admission and in hospital before elective PCI; n = 110), and both aspirin and dipyridamole (n = 32) [38]. They found a significant reduction in angiographic thrombus formation complicating the procedure with use of aspirin compared to no aspirin [38]. Another trial of 376 patients undergoing PCI randomized patients to aspirin–dipyridamole (330 mg–75 mg) or placebo. The aspirin–dipyridamole combination was given three times daily starting 24 h prior to PCI, eight hours prior to the procedure oral dipyridamole was replaced with intravenous dipyridamole at a dosage of 10 mg per hour for 24 h, and oral aspirin was continued. Sixteen hours after the procedure aspirin–dipyridamole (330 mg–75 mg) was continued. After 7 months of follow up, there was a significant reduction in transmural MI (defined as appearance of new Q waves) in the aspirin–dipyridamole compared with placebo group (1.6% vs. 6.9%; *p* = 0.011) [39].

Animal studies have suggested that aspirin attenuates the infarct size-limiting effects of statins, ticagrelor, opiates, and ischemic postconditioning [6]. However, there is a gap in the literature—possibly due to ethical concerns—on the effect of aspirin loading before PCI on attenuating the protection effects of statins, ticagrelor, opiates, and ischemic conditioning. A study conducted in rats by Birnbaum and colleagues found that when aspirin was given 15 min after reperfusion, the infarct size-limiting effect of pretreatment with atorvastatin was maintained, whereas when given before reperfusion the infarct-size limiting effect of atorvastatin was blocked [40]. We suggest that aspirin loading in patients with ACS, including STEMI, and its impact on infarct size and outcomes in combination with other beneficial therapies requires further investigation. Specifically, it should be studied whether aspirin loading is needed at all and if so, what is the optimal timing: immediately after the first encounter with the medical team or only after reperfusion, as is the current practice for P2Y12 inhibitors loading in patients with ACS.

## 2. Secondary Prevention

### 2.1. Long Term Treatment after STEMI and NSTE-ACS

The AHA/ACC 2014 NSTEMI-ACS guidelines recommended that patients be given aspirin maintenance dose of 81 to 325 mg per day (COR 1, LOE A) [2]. Similarly, 2023 ESC Guidelines for the management of acute coronary syndromes recommended maintenance dose of 75–100 mg once a day for long-term treatment [19]. These guidelines have been supported by several trials for aspirin maintenance therapy predating newer revascularization strategies. The Canadian Multicenter trial enrolled 555 unstable angina patients, given either aspirin (325 mg four times a day), sulfinpyrazone, combined aspirin and sulfinpyrazone, or a placebo. Aspirin (with or without sulfinpyrazone) decreased the composite of cardiac mortality or nonfatal MI and all-cause mortality compared to the non-aspirin group after a mean follow up of 18 months [41]. The Veterans Administration Cooperative Study recruited 1266 men with unstable angina within 48 h of admission who were randomized to aspirin 324 mg (n = 625) or placebo (n = 641). The study found lower risk of incidence of composite of mortality or MI (presence of creatine kinase MB or pathologic Q-wave changes) in patients with aspirin (5% aspirin vs. 10.1% placebo; *p* < 0.001) after 12 weeks of follow up [42].

A patient level meta-analysis of 16 trials (6 primary prevention and 10 secondary prevention trials) comparing aspirin to placebo found a significantly lower risk of coronary event (risk ratio, 0.80; 95% CI 0.73–0.88; event rate 4.30% vs. 5.30% per year; *p* < 0.001), ischemic stroke (risk ratio, 0.78; 95% CI 0.61–0.99; event rate 0.61% vs. 0.77% per year; *p* = 0.04), or serious vascular event (risk ratio, 0.81; 95% CI 0.75–0.87; event rate 6.69% vs. 8.19% per year; *p* < 0.001) [43]. It is important to note that these secondary outcomes were based on 16 trials, of which only 6 were post-MI trials. These six post-MI trials were conducted in the 1970s and 1980s, without the use of statins, P2Y12 inhibitors, or revascularization procedures.

### 2.2. Long-Term Aspirin Use after PCI

According to the current guidelines, aspirin should be continued indefinitely as a single antiplatelet therapy after completion of dual antiplatelet therapy (DAPT) for PCI [44,45].

P2Y12 inhibitors (clopidogrel or ticagrelor) have been shown to be superior when compared with aspirin as a single antiplatelet therapy after completion of dual antiplatelet therapies, in terms of both efficacy and safety, as depicted in Table 3 [17,18,46,47]. The Harmonizing Optimal Strategy for Treatment of Coronary Artery Diseases-Extended Antiplatelet Monotherapy (HOST-EXAM) trial included 5438 patients from South Korea (mean age 63.5 years, 74.5% males, almost one-third diabetic patients) who were randomized to receive clopidogrel 75 mg daily or aspirin 100 mg daily after completion of 6 to 18 months of DAPT following drug-eluting stent placement [ACS 72.1% (STEMI 17.2%, NSTEMI 19.4%, unstable angina 35.6%) or stable CAD] [18]. There was a significantly lower risk of all-cause mortality, MI, stroke, readmission due to ACS, or major bleeding (the primary end point) (hazard ratio 0.73; 95% CI 0.59 to 0.90; absolute risk reduction 2.0%; 95% CI 0.6 to 3.3; NNT = 51) and of major bleeding after a mean follow up of 2 years, (Table 4). These results must be interpreted with caution, as genetic and phenotypic testing for clopidogrel resistance were not considered, and this trial was conducted in East Asian patients who have a 50–60% prevalence of loss-of-function mutation of CYP2C19 leading to limited impact of clopidogrel on platelet function [48]. The GLOBAL LEADERS trial recruited patients from all over the world and had a large sample size (15,968 patients randomized in 1:1 ratio [N = 7980: 7988], 46.9% with ACS [STEMI 13.1%, NSTEMI 21.1%, unstable angina 12.7%] and 53.1% with stable CAD. It compared ticagrelor to aspirin after DAPT was given for one month following drug-eluting stenting [17]. After one month of DAPT, patients took ticagrelor alone for 23 months or aspirin plus clopidogrel or ticagrelor for 12 months, followed by aspirin monotherapy for 12 months. The study showed a non-significant trend to lower risk of the composite primary endpoint of all-cause mortality or new Q-wave MI (risk ratio, 0.87; 95% CI 0.75–1.01; *p* = 0.073) as well as stroke (risk ratio, 0.98; 95% CI 0.72–1.33), all-cause mortality (risk ratio, 0.88; 95% CI 0.74–1.06), and bleeding (risk ratio, 0.95; 95% CI 0.76–1.18) with ticagrelor compared to aspirin group (Table 4). It must be kept in mind that this trial was open label, leading to a potential for reporting bias and misclassification.

A network meta-analysis by Ando et al. assessed the effects of a P2Y12 inhibitor or aspirin following DAPT after PCI. The analysis included 19 studies with a total of 73,126 patients and concluded that aspirin is associated with a higher risk of developing MI (risk ratio, 1.32; 95% CI 1.08–1.62) compared to a P2Y12 inhibitor and a trend to higher risk of bleeding (risk ratio, 1.12; 95% CI 0.82–1.53) and stroke (risk ratio, 1.30; 95% CI 0.89–1.90) [49]. There is a growing body of evidence in favor of P2Y12 inhibitors as a single antiplatelet therapy compared with aspirin in terms of efficacy, safety, and net clinical benefit. Based on current data, a P2Y12 appears as effective as aspirin, and possibly more so, as a single antiplatelet agent.

**Table 4 jcm-12-07534-t004:** P2Y12 inhibitors compared to aspirin after PCI with DES.

Author (Year) Trial		HOST-EXAM (2021) [18]	TICO (2020) [50]	TWILIGHT (2019) [51]	SMART-CHOICE (2019) [46]	STOP-DAPT-2 (2019) [47]	GLOBAL LEADERS (2018) [17]
Geographical location		South Korea	Korea	International	Korea	Japan	International
No. of patients		5438	3056	7119	2993	3045	15,968
Male (%)		74.5	80	76.1	73.4	76.7	76.7
Age (yrs)		63.5	61	65.2	64.5	68.6	64.5
Diabetes (%)		34.2	27.3	36.8	37.5	38.1	25.3
Smoker (%)		20.7	37.4	21.8	26.4	23.3	26.1
Dyslipidemia (%)		69.3	60.4	60.4	45.2	73.7	67.4
Chronic Kidney Disease/impaired renal function (%)		12.7	20.3	16.8	3.24	5.5	13.6
Previous myocardial infarction (%)		16.0	3.70	28.7	4.24	13.3	23.2
Previous cerebrovascular accident (%)		4.7	4.12	NR	6.7	6.1	2.6
Indications for PCI	Stable angina/stable CAD	25.5	0	35.2	41.8	61.1	53.1
	Unstable angina	35.6	30.3	35.0	32.0	13.4	12.7
	NSTEMI	19.4	33.6	29.8	15.7	5.9	21.1
	STEMI	17.2	36.1	0	10.5	18.4	13.1
Follow up time (years)		2	1	1	1	1	2
DAPT duration		6–18 months in both arms	3-month DAPT followed by ticagrelor monotherapy vs. 12-month DAPT	3-month DAPT followed by P2Y12 inhibitors monotherapy vs. 12-month DAPT	3-month DAPT followed by P2Y12 inhibitors monotherapy vs. 12-month DAPT	1-month DAPT followed by clopidogrel o compared withstandard 12-month DAPT	Ticagrelor plus aspirin for 1 month, followed by ticagrelor monotherapy for 23 months vs. aspirin plus clopidogrel or ticagrelor for 12 months, followed by aspirin monotherapy for 12 months
Secondary prevention type		Dual antiplatelet therapy without clinical events for 6–18 months after percutaneous coronary intervention with DES	Acute coronary syndrome patients treated with drug-eluting stents	Dual antiplatelet therapy after percutaneous coronary intervention (PCI)	P2Y12 inhibitor monotherapy short-duration dual antiplatelettherapy (DAPT) vs. standard DAPT in patients undergoing percutaneous coronary intervention (PCI)	1 month of DAPT compared withstandard 12 months of DAPT	Percutaneous coronary intervention using DES for stable coronary artery disease or acute coronary syndromes
P2Y12 vs. Aspirin dose		Clopidogrel 75 mg vs. aspirin 100 mg	Aspirin 300 mg loading followed by 100 mg daily. Ticagrelor 180 mg loading followed by 90 mg daily	Aspirin 81 to 100 mg daily, ticagrelor 90 mg twice daily	Aspirin 100 mg once daily plus clopidogrel 75 mg once daily or prasugrel 10 mg once daily or ticagrelor 90 mg twice daily for 3 months in both groups	81 to 200mg/d, and clopidogrel,75 mg/d, or aspirin, 81 to 200 mg/d, and prasugrel 3.75 mg/d	Ticagrelor 90 mg twice daily vs. aspirin 75–100 mg daily
Outcomes	Myocardial infarction	HR = 0.65 (0.36 to 1.17)	HR = 0.55 (0.20 to 1.48)	HR = 1.0 (0.75 to 1.33)	HR = 0.66 (0.31 to 1.40)	HR = 1.19 (0.54 to 2.67)	RR = 1.0 (0.84 to 1.19)
	Stroke	HR = 0.42 (0.24 to 0.73)	HR = 0.73 (0.29 to 1.81)	HR = 2.0 (0.86 to 4.67)	HR = (2.23 (0.78 to 6.43)	HR = 0.50 (0.22 to 1.18)	RR = 0.98 (0.72 to 1.33)
	CV mortality	HR = 1.37 (0.69 to 2.73)	NR	HR = 0.70 (0.43 to 1.16)	HR = 0.86 (0.38 to 1.91)	HR = 0.83 (0.34 to 1.99)	NR
	All-cause mortality	HR = 1.43 (0.93 to 2.19)	HR = 0.70 (0.37 to 1.32)	HR = 0.75 (0.48 to 1.18)	HR = 1.18 (0.63 to 2.21)	HR = 1.18 (0.63 to 2.21)	RR = 0.88 (0.74 to 1.06)
	Bleeding	HR = 0.63 (0.41 to 0.97)	HR = 0.56 (0.34 to 0.91)	HR = 0.56 (0.45 to 0.68)	HR = 0.87 (0.40 to 1.88)	HR = 0.19 (0.05 to 0.65)	RR = 0.95 (0.76 to 1.18)

Abbreviations—CAD: coronary artery disease, DAPT: dual antiplatelet; DES: drug-eluting stents; HR: hazard ratio; NR: not reported; PCI: percutaneous coronary intervention; RRR: relative risk reductions; RR: risk ratio.

### 2.3. DAPT Duration

The 2023 ESC acute coronary syndrome guidelines recommended that single antiplatelet therapy (preferably with a P2Y12 receptor inhibitor) should be considered for patients who are event free after 3–6 months of DAPT and who are not at high ischemic risk and that in high bleeding risk patients, aspirin or P2Y12 receptor inhibitor monotherapy may be considered after 1 month of DAPT (COR 2a, LOE A) [19]. The 2021 ACC/AHA/Society for Cardiovascular Angiography and Interventions guidelines for coronary artery revascularization recommended that shorter-duration DAPT (1–3 months) is reasonable, with a subsequent transition to P2Y12 inhibitor monotherapy to reduce the risk of bleeding events in selected patients undergoing PCI (COR 2a, LOE A) [37]. The 2023 AHA/ACC/ACCP/ASPC/NLA/PCNA Guideline for the Management of Patients With Chronic Coronary Disease recommended DAPT consisting of aspirin and clopidogrel for 6 months post PCI followed by single antiplatelet therapy, which is indicated to reduce MACE and bleeding events (COR 1, LOE A) [33]. Similarly, these guidelines also recommended that in select patients with chronic coronary disease treated with PCI and a drug-eluting stent (DES) who have completed a 1- to 3-month course of DAPT, P2Y12 inhibitor monotherapy for at least 12 months is reasonable to reduce bleeding risk (COR 2a, LOE A) [33].

Recently, three shorter duration DAPT trials after drug-eluting stenting have used P2Y12 inhibitors for maintenance antiplatelet therapy [46,47,51]. The Comparison Between P2Y12 inhibitor Monotherapy vs. Dual Antiplatelet Therapy in Patients Undergoing Implantation of Coronary Drug-Eluting Stents (SMART CHOICE) trial, conducted in Korea, compared 3 months of DAPT followed by P2Y12 inhibitors (clopidogrel, prasugrel, or ticagrelor) to 12 months of DAPT. In contrast, the ShorT and OPtimal Duration of Dual AntiPlatelet Therapy- Study (STOP-DAPT-2) trial, conducted in Japan, compared 1 month of DAPT followed by P2Y12 inhibitors (clopidogrel or prasugrel) to 12 months of DAPT. Both trials recruited around 3000 patients with a similar age and gender distribution. SMART-CHOICE trial, a non-inferiority trial, showed that a short DAPT followed by P2Y12 inhibitors (clopidogrel, prasugrel, or ticagrelor) was non-inferior to standard DAPT therapy for a composite of all-cause death, MI, or stroke (primary end point) compared to 12 months of DAPT (risk difference, 0.4%; 1-sided 95% CI, –∞%–1.3%; *p* = 0.007 for noninferiority). STOP-DAPT-2 found a lower risk (hazard ratio, 0.64; 95% CI 0.42–0.98) of the composite of cardiovascular death, MI, definite stent thrombosis, ischemic or hemorrhagic stroke, or TIMI major or minor bleeding (primary end point) for 1 month of P2Y12 inhibitor (clopidogrel or prasugrel) compared to standard duration DAPT (12 months) which was non-inferior (*p* < 0.001) and superior (*p* = 0.04) as shown in Table 4. The Ticagrelor With Aspirin or Alone in High-Risk Patients After Coronary Intervention (TWILIGHT) study recruited 9006 patients with high risk of bleeding or an ischemic event and compared 3 months of DAPT (aspirin and ticagrelor combined) followed by ticagrelor monotherapy to 12 months of DAPT (combined aspirin and ticagrelor) [51]. The study found a lower risk of composite end point of death from any cause, nonfatal MI, or nonfatal stroke (hazard ratio, 0.56; 95% CI 0.45–0.68; *p* < 0.001) and a lower risk of bleeding (hazard ratio, 0.49; 95% CI 0.33–0.74; *p* < 0.001) with short-term DAPT followed by ticagrelor monotherapy [51]. Less than one month of DAPT therapy is also feasible, as evidenced by the T-PASS trial [52], which was a multi-center, open-labelled, non-inferiority trial performed in Korea. It enrolled 2850 patients (mean age 61 years, 83% males, 40% STEMI) with ACS and underwent drug-eluting stent randomized in a 1:1 ratio with either ticagrelor monotherapy (90mg twice daily) after less than 1 month of DAPT (median 16 days) or 12 months of DAPT (ticagrelor and aspirin) and found a lower primary end point (defined as net clinical benefit as a composite of all-cause death, myocardial infarction, definite or probable stent thrombosis, stroke, and major bleeding at 1 year after the index procedure) with less than 1 month of DAPT compared to 12 months of DAPT (2.8% vs. 5.2%, HR = 0.54; 95 CI: 0.37–0.80; *p* < 0.001), which is both superior and non-inferior.

In a recent proof of concept MACT trial (Mono Antiplatelet and Colchicine Therapy), which enrolled 200 patients with ACS treated with drug-eluting stents in which aspirin was discontinued on the day after PCI and treated with colchicine in addition to ticagrelor or prasugrel maintenance therapy, found 2 patients (1%) with stent thrombosis (primary outcomes) at 3 months of follow up as well as favorable platelet function and inflammatory profiles [53]. This single-arm study showed a low rate of stent thrombosis but was limited by the small number of patients studied; further studies and larger trials are required to further test this pilot trial.

A pooled patient-level meta-analysis of 22,941 patients from five RCTs compared short-term DAPT (one to three months duration) followed by P2Y12 inhibitor monotherapy to standard DAPT therapy after complex PCI (defined as one of the following: 3 vessels treated, ≥3 stents implanted, ≥3 lesions treated, bifurcation with 2 stents implanted, total stent length >60 mm, or chronic total occlusion) and found similar fatal and ischemic events and significantly lower risk of major bleeding with short-term DAPT [54].

Thus, it seems that after a short duration of DAPT (1–3 months) there is likely no need to continue aspirin if P2Y12 inhibitor monotherapy is used, particularly in high bleeding risk patients. Based on the aforementioned trials, the 2023 ESC guidelines for the management of acute coronary syndromes recommend that single antiplatelet therapy (preferably with a P2Y12 receptor inhibitor) should be considered for patients who are event free after 3–6 months of DAPT and who are not at a high ischemic risk. The remaining clinically relevant questions include the following: Do we need the initial DAPT treatment phase? Can (or should) we omit the aspirin at all or at least shortly after PCI? These questions have not been investigated extensively in clinical trials.

### 2.4. Aspirin Use in Patients Undergoing CABG

The ACC/AHA 2021 guidelines for coronary artery revascularization recommended that, to reduce ischemic events, patients who are currently taking aspirin should continue to take aspirin until surgery is performed (COR 1, LOE B-R) [37]. However, these guidelines recommend against the initiation of aspirin prior to CABG in patients who are not taking aspirin (COR 3, LOE B-R) [37]. Aspirin should be administered within 6 h after CABG in doses of 100 to 325 mg daily, followed by indefinite maintenance therapy to reduce risk of graft occlusion and adverse cardiac events (COR 1, LOE A) [37,55].

One of the first double-blind placebo-controlled trials studied 407 patients randomized to either dipyridamole (initiated two days prior to CABG) and aspirin (initiated 7 h after CABG) or to placebo. The study reported a significantly lower venous graft occlusion at one month after CABG with dipyridamole and aspirin combined compared to placebo (3% vs. 10%, *p* < 0.001) [56].

### 2.5. Contribution of P2Y12 Inhibitor Use

#### 2.5.1. Primary Prevention

There is paucity of data on the role of P2Y12 inhibitors versus placebo or aspirin for primary prevention. In the quest for better and safer therapy, there is a need to assess the safety and efficacy of P2Y12 inhibitors use for primary prevention in comparison with aspirin.

#### 2.5.2. Secondary Prevention

##### Loading for STEMI, NSTEMI or Elective PCI

The ACC/AHA 2021 guidelines for coronary artery revascularization and ESC 2023 guidelines for management of acute coronary syndrome recommends a loading dose of a P2Y12 inhibitor for patients undergoing PCI with stenting (COR I, LOE A) [19,37]. This recommendation came after Clopidogrel for the Reduction of Events During Observation (CREDO) trial which included 2116 patients (13.7% recent MI, 52.8% unstable angina, 2.8% stable angina) scheduled to undergo elective PCI. Patients were randomized to clopidogrel loading dose of 300 mg (n = 1053) or placebo (n = 1063) 3 to 24 h prior to PCI in addition to aspirin [57]. Clopidogrel did not reduce the risk of 28-day composite of all-cause mortality, MI, or urgent target vessel revascularization compared to a placebo; however, a post hoc analysis showed that patients who received clopidogrel at least 6 h before PCI experienced relative risk reduction of 38.6% (CI: −1.6% to 62.9%; *p* = 0.051) compared to placebo, supporting that adequate dose and duration of clopidogrel is important for the full antiplatelet effect. Subsequently, the PCI-Clopidogrel as Adjunctive Reperfusion Therapy (PCI-CLARITY) trial enrolled 1863 patients who underwent PCI for STEMI and received aspirin. Patients were randomized to either clopidogrel (300 mg loading dose, then 75 mg once daily) or placebo initiated with fibrinolysis and given until coronary angiography, which was performed 2 to 8 days after initiation of the study drug, at the discretion of physician. The study found significant reduction in the composite incidence of cardiovascular death, MI, or stroke following PCI with clopidogrel (3.6% vs. 6.2%; *p* = 0.008) [58]. Finally, a clopidogrel loading dose of 600 mg was proven better than 300 mg. It was supported by the Antiplatelet therapy for Reduction of Myocardial Damage during Angioplasty (ARMYDA-2) trial which enrolled 255 patients (NSTE ACS 25%, stable angina 75%) and were randomized to 600 mg clopidogrel dose (n = 126) or 300 mg clopidogrel dose (n = 129) 4–8 h pre-PCI [59]. There was significantly lower risk of 28-day composite of all-cause mortality, MI, or target vessel revascularization with the high-dose versus low-dose clopidogrel (4% vs. 12%; *p* = 0.041) with similar safety endpoints (no major bleeding or requiring transfusion for either group). It is notable that the aforementioned composite outcome was driven by incidence of MI (defined as rise in creatine kinase MB, troponin I, and myoglobin levels); as such, its clinical meaning is less clear. In a recent SHORT-DAPT-3 trial which enrolled 5966 patients with elective PCI, high risk of bleeding or ACS conducted in Korea showed that prasugrel monotherapy at a dose of 3.75 mg/day was not superior to DAPT with aspirin 81–100 mg/day and prasugrel 3.75 mg/day for bleeding events (BARC 3 or 5) but has higher primary events (defined as composite of cardiovascular death, MI, definite stent thrombosis, or stroke) at one month of follow up [60]. However, it must be kept in mind that the dose of prasugrel was lower than the standard dose used in clinical practice (5 mg) and the duration of follow up was only one month [60].

According to ESC 2023 guidelines for the management of acute coronary syndrome, pre-treatment (P2Y12 administration before angiography when coronary anatomy is unknown) with P2Y12 inhibitor may be considered in STEMI and NSTE-ACS with delayed invasive angiography (>24 h), but is not recommended for NSTE-ACS with early invasive angiography (<24 h) strategy [19]. Pre-treatment in STEMI recommendations came from the ATLANTIC trial in patients with STEMI. Administration of Ticagrelor in the Cath Lab or in the Ambulance for New ST Elevation Myocardial Infarction to Open the Coronary Artery which randomized 1862 STEMI patients to ticagrelor loading dose administration during transfer to PCI center or immediately before angiography and found no difference in co-primary end points were the proportion of patients who did not have a 70% or greater resolution of ST-segment elevation before percutaneous coronary intervention (PCI) and the proportion of patients who did not have Thrombolysis in Myocardial Infarction flow grade 3 in the infarct-related artery at initial angiography between two groups, but there was lower risk of secondary end point definite stent thrombosis with pretreatment at 24 h (0% vs. 0.8%; *p* = 0.008) and 30 days (0.2% vs. 1.2%; *p* = 0.02) [61]. Pretreatment is not recommended in NSTE-ACS with early invasive angiography (<24 h) strategy which is based on A Comparison of Prasugrel at the Time of Percutaneous Coronary Intervention or as Pretreatment at the Time of Diagnosis in Patients with Non-ST Elevation Myocardial Infarction (ACCOST) trial [62]. In a select group of patients who cannot tolerate oral medications, cangrelor can be administered. The ESC 2023 guidelines for management of acute coronary syndrome states that in P2Y12 receptor inhibitor-naive patients undergoing PCI, cangrelor may be considered [19]. This came after CHAMPION PCI trial which randomized 8716 patients (5.2% with stable angina,35.4% unstable angina, and 59.4% with non-STE myocardial infarction) in 1:1 ratio to cangrelor and clopidogrel group administration before PCI and found cangrelor was not superior to clopidogrel (OR = 1.05 with 95% CI: 0.88 to 1.24; *p* = 0.59) in reduction of primary outcome (which was a composite of death from any cause, myocardial infarction, or ischemia-driven revascularization at 48 h) [63]. Later CHAMPION PHEONIX trials assigned 11,145 patients undergoing either urgent or elective PCI randomized in 1:1 ratio to either cangrelor bolus and infusion or clopidogrel loading dose of 300 or 600 mg found lower risk of primary efficacy end point with cangrelor (adjusted OR = 0.78 with 95% CI 0.66 to 0.93; *p* = 0.005) with no difference in bleeding risk compared to clopidogrel (composite of death, myocardial infarction, ischemia-driven revascularization, or stent thrombosis at 48 h after randomization) [64]. P2Y12 alone (with adequate dose) without aspirin loading has not been extensively studied. Can it be that the additional benefits of aspirin and P2Y12 inhibitors combination are mainly due to the P2Y12 inhibitors and can be maintained even without concomitant use of aspirin? Further investigations are required to evaluate this hypothesis.

##### Long-Term Treatment for Secondary Prevention

In general, P2Y12 inhibitors have shown better efficacy and safety profiles than aspirin in ACS trials. The Stent Anticoagulation Restenosis study (STARS) included 1965 patients randomized to aspirin monotherapy (n = 557 patients), aspirin and warfarin (n = 550 patients), and aspirin and ticlopidine (n = 546) after coronary artery stenting and evaluated for 30-day primary composite endpoint of death, revascularization of the target lesion, angiographically evident thrombosis, or MI [65]. The study found lower risk of the primary end point with aspirin and ticlopidine combination therapy compared to aspirin alone (RR = 0.15, 95% CI 0.05–0.43; *p* < 0.001) but higher risk of hemorrhagic complications (RR = 3.06, 95% CI 1.57–5.97; *p* = 0.002). Ticlopidine, a first generation P2Y12 inhibitor, has been subsequently replaced by second- and third-generation agents.

Clopidogrel’s use in clinical practice for secondary prevention became prevalent after clopidogrel versus aspirin in patients at risk of ischemic events (CAPRIE) trial which included 19,185 patients [10]. This trial recruited atherosclerotic vascular disease patients [either recent ischemic stroke (33.5%), recent MI (32.8%), or symptomatic peripheral arterial disease (33.6%)] with at least 1-year follow up. Clopidogrel was associated with a lower risk of the composite primary endpoint of ischemic stroke, MI, or vascular death (relative risk reduction, 8.7%; 95% CI, 0.3–16.5) with event rate 5.32% vs. 5.82 and absolute difference 0.51% lower with clopidogrel which was statistically significant (*p* = 0.042) (Table 3). A meta-analysis of nine randomized clinical trials (61,623 patients) comparing P2Y12 inhibitors (five trials with clopidogrel and four trials with ticagrelor) to aspirin for secondary prevention of cardiovascular events found a significantly lower MACE risk (risk ratio, 0.89; 95% CI 0.84–0.95) and MI risk (risk ratio, 0.81; 95% CI 0.71–0.92), as well as a non-significant lower risk of stroke (risk ratio, 0.85; 95% CI 0.73–1.01) and major bleeding (risk ratio, 0.94; 95% CI 0.72–1.22) with P2Y12 inhibitors [66]. It is important to consider that this analysis included trials requiring antiplatelet for secondary prevention including stroke, TIA, and peripheral arterial disease as well as for CAD.

##### Long-Term P2Y12 Use after CABG

Although aspirin seems to be beneficial after CABG compared to placebo, recent data has suggested equal efficacy of ticagrelor. The Ticagrelor With Aspirin for Prevention of Vascular Events in Patients Undergoing coronary artery bypass graft (TiCAB) trial (1859 patients) included patients with stable CAD or ACS requiring CABG and randomized patients to ticagrelor 90 mg twice daily or aspirin 100 mg once daily in 1:1 ratio [13]. This trial showed no significant difference in ticagrelor compared to aspirin (HR = 1.19; 95% CI:0.87–1.62; *p* = 0.28) for the composite of cardiovascular death, MI, repeat revascularization, and stroke between groups 12 months after CABG. Another trial, the Dual Ticagrelor Plus Aspirin Antiplatelet Strategy After Coronary Artery Bypass Grafting (DACAB) trial randomized 500 patients to ticagrelor and aspirin, ticagrelor monotherapy, and aspirin monotherapy in 1:1:1 ratio. Aspirin plus ticagrelor showed significantly higher patency rate compared with aspirin alone (absolute risk difference, 12.2%; 95% CI, 5.2–19.2%; *p*  <  0.001; 88.2% vs. 76.5%), but no significant difference between ticagrelor and aspirin (risk difference, 6.3%, 95% CI, −1.1–13.7%; *p* = 0.10; 82.8% vs. 76.5%) for saphenous venous graft patency one year after CABG [14]. There was no significant difference in major bleeding events (total number of patients five; three in ticagrelor plus aspirin, two in ticagrelor alone, zero in aspirin alone). Although there is no difference between aspirin and P2Y12 inhibitors, the use of DAPT is recommended is efficacious compared to aforementioned monotherapy, as reflected in the 2016 ACC/AHA Guideline Focused Update on the duration of DAPT in patients with CAD guidelines—patients with ACS treated with CABG should be continued on DAPT for 12 months after ACS (class I and LOE C-LD) [45]. The findings of this study are summarized in Figure 1.

## 3. Conclusions

The recommendations for aspirin for primary prevention have been recently downgraded in view of current scientific evidence. The role of aspirin loading in patients with ACS and those undergoing PCI has never been studied separately, although it is considered a COR I recommendation by current guidelines. Aspirin has been shown to be superior to placebo for long-term maintenance therapy, but has lower efficacy compared to P2Y12 inhibitors for secondary prevention after stent placement for STEMI, NSTEMI, or stable CAD. The current evidence strongly suggests that P2Y12 inhibitor therapy alone after a short DAPT duration is safe and effective for patients after PCI. Remaining clinically important questions include the feasibility of foregoing loading dose of aspirin or at least delay aspirin loading not to block the preconditioning effects of additional therapies such as statins, and the need for DAPT with aspirin for the first 1 to 3 months after PCI.

## Figures and Tables

**Figure 1 jcm-12-07534-f001:**
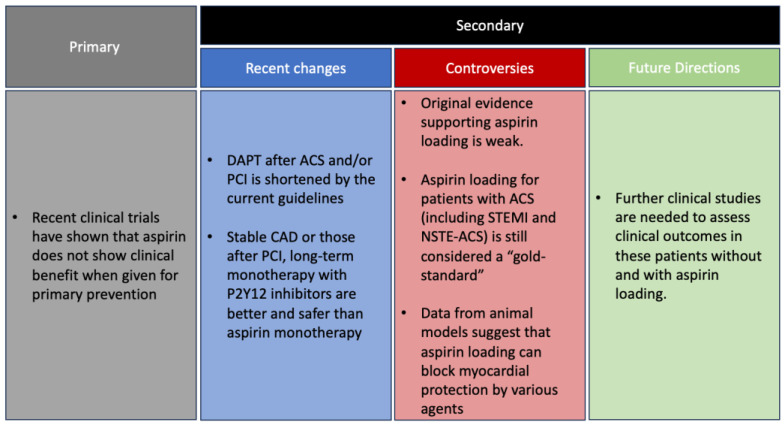
Primary and secondary prevention of aspirin, controversies, and future directions.

**Table 1 jcm-12-07534-t001:** Diabetes and non-diabetes trials evaluating aspirin for primary prevention.

Author (Year) Trial	Study Period	No. of Patients	Male (%)	Age (yrs)	Diabetes (%)	Aspirin Dose/Day	Outcomes in RR or HR			
							Myocardial Infarction	Stroke	CV Mortality	All-Cause Mortality
**Non-diabetes**										
Peto R (1988) [20]	1978–1984	5139	100	61	2	500 mg	NR	NR	NR	NR
Ongoing Physician Health Study (1989) [21]	1982–1988	22,071	NR	NR	2	325 mg	RR = 0.56 (0.45–0.70)	RR = 1.22 (0.93–1.60)	RR = 0.96 (0.60–1.54)	RR = 0.96 (0.80–1.14)
Hansson (1998) HOT trial [22]	1992–1997	18,790	53	61	8	75 mg	RR = 0.85 (0.69–1.05)	RR = 0.98 (0.78–1.24)	RR = 0.95 (0.75–1.20)	RR = 0.93 (0.79–1.09)
Thrombosis prevention trial (1998) [23]	1984–1997	5499	100	57	2	75 mg	RR = 0.80 (0.64–0.99)	RR = 0.98 (0.65–1.47)	RR = 1.26 (0.93–1.69)	RR = 1.03 (0.80–1.32)
Primary prevention project (2001) [24]	1994–1998	4495	42	64	17	100 mg	RR = 0.69 (0.38–1.23)	RR = 0.67 (0.36–1.27)	RR = 0.56 (0.31–0.99)	RR = 0.81 (0.58–1.13)
Ridker P (2005) Women’s health study [25]	1992–2004	39,876	0	54	3	100 mg	RR = 1.02 (0.84–1.25)	RR = 0.83 (0.69–0.99)	RR = 0.95 (0.74–1.22)	RR = 0.95 (0.85–1.06)
Fowkes (2010) Aspirin for Asymptomatic Atherosclerosis trial [26]	1998–2008	3350	28	62	3	100 mg	NR	NR	NR	HR = 0.95 (0.77–1.16)
Ikeda Y (2014) Japanese Primary Prevention Project (JPPP) [27]	2002–2008	2539	55	65	34	100 mg	HR = 0.53 (0.31–0.91)	HR = 1.04 (0.80–1.34)	HR = 1.03 (0.71–1.48)	HR = 0.99 (0.85–1.17)
Gaziano (2018) ARRIVE [7]	2007–2016	12,546	70	64	0	100 mg	HR = 0.85 (0.64–1.11)	1.12 (0.80–1.55)	HR = 0.97 (0.62–1.52)	HR = 0.99 (0.80–1.24)
McNeil (2018) ASPREE [9]	2010–2014	19,114	44	74	11	100 mg	HR = 0.93 (0.76–1.15)	HR = 0.95 (0.83–1.08)	NR	NR
**Diabetes**										
Ogawa (2008) Japanese Primary Prevention of Atherosclerosis with Aspirin for diabetes [28]	2002–2008	2539	55	65	100	81 or 100 mg	NR	HR = 0.84 (0.53–1.32)	NR	NR
Belch (2008) Prevention of progression of arterial disease and diabetes (POPADAD) [29]	1997–2006	1276	44	60	100	100 mg	HR = 0.98 (0.68–1.43)	HR = 0.71 (0.44–1.14)	HR = 1.35 (0.81–2.25)	HR = 0.93 (0.71–1.24)
ASCEND (2018) [8]	2005–2017	15,480	63	63	100	100 mg	HR = 0.98 (0.80–1.19)	0.88 (0.73–1.06)	NR	NR

Abbreviations—HR: hazard ratio, NR: not reported, RR: risk ratio.

**Table 2 jcm-12-07534-t002:** Society guidelines and scientific statements for primary and secondary prevention with aspirin.

Society	Guideline	Class of Recommendation	Level of Evidence
**Primary prevention**			
American College of Cardiology/American Heart Association guidelines for primary prevention of cardiovascular disease 2019 [3]	Low-dose aspirin 75–100 mg for primary prevention between ages 40 and 70 years with higher risk of atherosclerotic cardiovascular disease and not at increased risk of bleeding	IIb	A
Preventive Services Task Force Recommendation Statement 2022 [5]	Aspirin as primary prevention for patient between 40 and 59 years who have a 10% or greater risk of cardiovascular disease who are not at increased risk of bleeding	-	C
**Secondary prevention**			
2013 ACCF/AHA Guideline for the Management ofST-Elevation Myocardial Infarction [1]	Aspirin 162 to 325 mg should be given before primary PCI	I	B
	After PCI, aspirin should be continued indefinitely	I	A
2014 AHA/ACC Guideline for the Management of Patients With Non–ST-Elevation AcuteCoronary Syndromes [2]	Non–enteric-coated, chewable aspirin (162 mg to 325 mg) should be given to all patients with NSTE-ACS without contraindications as soon as possible after presentation, and a maintenance dose of aspirin (81 mg/d to 325 mg/d) should be continued indefinitely	I	A
	Aspirin maintenance dose continued indefinitely	I	A
	Patients already taking daily aspirin before PCI should take 81 mg to 325 mg non–enteric-coated aspirin before PCI	I	B
2020 ESC Guidelines for the management of acute coronary syndromes in patients presenting without persistent ST-segment elevation [4]	Aspirin is recommended for all patients without contraindications at an initial oral loading dose of 150 to 300 mg (or 75 to 250 mg i.v.), and at a maintenance dose of 75 to 100 mg daily for long-term treatment.	I	A
2015 Secondary prevention after coronary artery bypass graft surgery: a scientific statement from the American Heart Association [32]	Aspirin should be administered preoperatively and within 6 h after CABG in doses of 81 to 325 mg daily followed by indefinite maintenance therapy to reduce graft occlusion and adverse cardiac events	I	A
2023 AHA/ACC/ACCP/ASPC/NLA/PCNA guidelines for the management of patients with chronic coronary disease [33]	Dual antiplatelet therapy consisting of aspirin and clopidogrel for 6 months post PCI followed by single antiplatelet therapy is indicated to reduce MACE and bleeding events	I	A
	Patient with chronic coronary disease recommended PCI and a drug-eluting stent who completed a 1- to 3-month course of dual antiplatelet therapy, P2Y12 inhibitor monotherapy for at least 12 months is reasonable to reduce bleeding risk	IIa	A
2023 ESC Guidelines for the management of acute coronary syndromes [19]	P2Y12 inhibitor monotherapy may be considered as an alternative to aspirin monotherapy for long-term treatment	IIb	A
	In high bleeding risk patients, aspirin or P2Y12 receptor inhibitor monotherapy after 1 month of DAPT may be considered	IIb	A
	In patients who are event free after 3–6 months of DAPT and who are not at high ischemic risk, single antiplatelet therapy (preferably with a P2Y12 receptor inhibitor) should be considered	IIa	A

Abbreviations—CABG: coronary artery bypass graft; MACE: major adverse cardiovascular events; PCI: percutaneous coronary intervention.

**Table 3 jcm-12-07534-t003:** Ticagrelor and clopidogrel compared to aspirin for secondary prevention as single antiplatelet therapy without initial dual antiplatelet therapy.

Author (Year) Trial	Geographical Location	N	Male (%)	Age (yrs)	Follow up Time (yrs)	Secondary Prevention Type	P2Y12 vs. Aspirin Dose	Outcomes in RR or HR				
								MI	Stroke	CV Mortality	All-Cause Mortality	Bleeding
**Clopidogrel (C)**												
CAPRIE (1996) [10]	International	19,185	71.9	62.5	3	ASCVD (ischemic stroke, MI, or PAD)	ASA 325 mg vs. C 75 mg	NR	NR	NR	RRR = 2.2 (−9.9 to 12.9)	NR
CADET (2004) [12]	United Kingdom	184	80.9	62.6	0.5	Acute MI within the previous 3–7° days	C 75 mg vs. ASA 75 mg	NR	NR	NR	NR	NR
ASCET (2012) [11]	Norway	1001	78.2	62.4	2	SIHD	ASA 160 mg vs. C 75 mg	RR = 2.05 (0.62 to 6.80)	RR = 2.05 (0.37 to 11.17)	NR	NR	NR
**Ticagrelor (T)**												
SOCRATES (2016) [15]	International	13,199	58.4	65.8	0.25	Acute ischemic stroke or transient ischemic attack	T 90 mg twice daily vs. ASA 100 mg	HR = 1.20 (0.67 to 2.14)	HR = 0.86 (0.75 to 0.99)	HR = 1.18 (0.75 to 1.85)	HR = 1.18 (0.83 to 1.67)	HR = 0.83 (0.52 to 1.34)
DACAB (2018) [14]	China	332	82.8	63.6	1	Post-coronary artery bypass grafting	T 90 mg twice daily vs. ASA 100 mg	NR	NR	NR	NR	NR
TICAB (2019) [13]	International	1859	84.9	66.7	1	ACS or SIHD post- CABG	T90 mg twice daily vs. aspirin 100 mg	HR = 0.63 (0.36–1.12)	HR = 1.21 (0.70–2.08)	HR = 0.85 (0.38–1.89)	NR	HR = 1.02 (0.67–1.55)

Abbreviations—ASA: aspirin; ASCVD: atherosclerotic cardiovascular disease; C = clopidogrel; CV = cardiovascular; HR: hazard ratio; N = number of patients; NR: not reported; RRR: relative risk reductions; RR: risk ratio; SIHD: stable ischemic heart disease; T = ticagrelor.

## Data Availability

Not applicable.

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
