# Peer review of "Do We Still Need Aspirin in Coronary Artery Disease?"

_jcm, 2023, doi:10.3390/jcm12247534_

Round 1
Reviewer 1 Report
Comments and Suggestions for Authors
The Authors propose a bold review of the literature in a very huge field of interest. Despite the interesting references and tables reported, the Authors must better organize the paper by dividing primary prevention, secondary prevention, secondary prevention after PCI and post DAPT, DAPT duration.
Particularly DAPT duration and secondary prevention are mixed (ex.: GLOBAL LEADERS focus on DAPT duration, not secondary prevention)
In the chapter of DAPT duration (short vs long) must be cited the most important scores (ex: precise dapt) and their components.
In the chapter of antiplatelet loading dose, something about iv usage of ASA and Cangrelor should be noted
Author Response
Reviewer 1:
The Authors propose a bold review of the literature in a very huge field of interest. Despite the interesting references and tables reported, the Authors must better organize the paper by dividing primary prevention, secondary prevention, secondary prevention after PCI and post DAPT, DAPT duration.
Particularly DAPT duration and secondary prevention are mixed (ex.: GLOBAL LEADERS focus on DAPT duration, not secondary prevention)
In the chapter of DAPT duration (short vs long) must be cited the most important scores (ex: precise dapt) and their components.
Response:
The sections already divided into primary and secondary prevention.
We could not added precise DAPT score due to the following reasons: 1) DAPT score was not quoted in the trials or 2) all individual components used to calculate precise DAPT score were not available in the trials.
In the chapter of antiplatelet loading dose, something about iv usage of ASA and Cangrelor should be noted
Response:
Line 397-411
We added the following:
“According to ESC 2023 guidelines for management of acute coronary syndrome P2Y12 receptor inhibitor in naive patients undergoing PCI, cangrelor may be considered 32. This came after CHAMPION PCI trial which randomized 8716 patients (5.2% with stable angina,35.4% unstable angina, and 59.4% with non-STE myocardial infarction) in 1:1 ratio to cangrelor and clopidogrel group administration before PCI and found cangrelor was not superior to clopidogrel (OR = 1.05 with 95% CI: 0.88 to 1.24; p = 0.59) in reduction of primary outcome (which was a composite of death from any cause, myocardial infarction, or ischemia-driven revascularization at 48 hours) 62. Later CHAMPION PHEONIX trials assigned 11,145 patients undergoing either urgent or elective PCI randomized in 1:1 ratio to either cangrelor bolus and infusion or clopidogrel loading dose of 300 or 600 mg and found lower risk of primary efficacy end point with cangrelor (adjusted OR = 0.78 with 95% CI 0.66 to 0.93; p = 0.005) with no difference in bleeding risk compared to clopidogrel (composite of death, myocardial infarction, ischemia-driven revascularization, or stent thrombosis at 48 hours after randomization) 63.”
Reviewer 2 Report
Comments and Suggestions for Authors
I read with interest the Review by Muhammad Haisum Maqsood et al.
This is a Review on a contemporary question regarding the role of Aspirin in the modern era of Cardiology.
It provides a comprehensive overview of studies on Aspirin in primary and secondary prevention and in the context of PCI and CABG as a mode of revascularization.
In all, I believe that this is a nicely written Review that encapsulates pertinent studies results. Although, the content of this Review is already presented in recent Guidelines, I believe there is value to the readers interested in this topic.
Author Response
Reviewer 2:
I read with interest the Review by Muhammad Haisum Maqsood et al.
This is a Review on a contemporary question regarding the role of Aspirin in the modern era of Cardiology.
It provides a comprehensive overview of studies on Aspirin in primary and secondary prevention and in the context of PCI and CABG as a mode of revascularization.
In all, I believe that this is a nicely written Review that encapsulates pertinent studies results. Although, the content of this Review is already presented in recent Guidelines, I believe there is value to the readers interested in this topic.
Response:
We thank the reviewer for investing time in this great review.
Reviewer 3 Report
Comments and Suggestions for Authors
Muhammad Haisum Maqsood et al. present a review article with the aim to discuss the role of aspirin in light of available evidence for primary prevention, secondary prevention for stable coronary artery disease or acute coronary syndromes, and after percutaneous coronary intervention or coronary artery bypass revascularization. The topic is interesting and the manuscript well-written. The tables are well presented.
1) Update the Introduction by adding the latest ESC Guidelines on acute coronary syndrome: Byrne RA, et al.; ESC Scientific Document Group. 2023 ESC Guidelines for the management of acute coronary syndromes. Eur Heart J Acute Cardiovasc Care. 2023 Sep 22:zuad107. doi: 10.1093/ehjacc/zuad107.
2) Referring to the Introduction, could you add a figure on the current use of aspirin in primary and secondary prevention?
3) Prescribing Aspirin and P2Y12 inhibitors in older ages is challenging. Please, add a specific paragraph on this topic with the current evidence, recommendations and your suggestions on the management.
Please, add these reference to improve the new sentences:
- McNeil JJ, et al.; ASPREE Investigator Group. Effect of Aspirin on Cardiovascular Events and Bleeding in the Healthy Elderly. N Engl J Med. 2018 Oct 18;379(16):1509-1518. doi: 10.1056/NEJMoa1805819.
- Cacciatore S, et al. Management of Coronary Artery Disease in Older Adults: Recent Advances and Gaps in Evidence. J Clin Med. 2023 Aug 11;12(16):5233. doi: 10.3390/jcm12165233.
4) Please, add a specific paragraph on “Gaps in evidence and future directions” before the conclusions.
Author Response
Reviewer 3:
Muhammad Haisum Maqsood et al. present a review article with the aim to discuss the role of aspirin in light of available evidence for primary prevention, secondary prevention for stable coronary artery disease or acute coronary syndromes, and after percutaneous coronary intervention or coronary artery bypass revascularization. The topic is interesting and the manuscript well-written. The tables are well presented.
1) Update the Introduction by adding the latest ESC Guidelines on acute coronary syndrome: Byrne RA, et al.; ESC Scientific Document Group. 2023 ESC Guidelines for the management of acute coronary syndromes. Eur Heart J Acute Cardiovasc Care. 2023 Sep 22:zuad107. doi: 10.1093/ehjacc/zuad107.
Response:
Line 60-61
We added the following in the introduction:
“Further, European Society of Cardiology guidelines for acute coronary syndrome has recently been published. 19 In the light of new evidence,”
2) Referring to the Introduction, could you add a figure on the current use of aspirin in primary and secondary prevention?
Response:
Line 464-466
We added the following figure:
“The findings of this study are summarized in figure 1.
Figure 1. Primary and secondary prevention of aspirin, controversies, and future directions.
3) Prescribing Aspirin and P2Y12 inhibitors in older ages is challenging. Please, add a specific paragraph on this topic with the current evidence, recommendations and your suggestions on the management.
Please, add these reference to improve the new sentences:
- McNeil JJ, et al.; ASPREE Investigator Group. Effect of Aspirin on Cardiovascular Events and Bleeding in the Healthy Elderly. N Engl J Med. 2018 Oct 18;379(16):1509-1518. doi: 10.1056/NEJMoa1805819.
- Cacciatore S, et al. Management of Coronary Artery Disease in Older Adults: Recent Advances and Gaps in Evidence. J Clin Med. 2023 Aug 11;12(16):5233. doi: 10.3390/jcm12165233.
Response:
Line 90-91
We already included ASPREE trial. We added the following:
“These risks of bleeding exponentiates with advanced age and frailty 8,30.”
4) Please, add a specific paragraph on “Gaps in evidence and future directions” before the conclusions.
Response:
Line 28-44
We added the following before the introduction:
“What is New: For years there was a recommendation for aspirin for primary prevention in individuals at risk for and for patients with established coronary artery disease. Those recommendations were based on studies conducted before the introduction of other therapies that significantly affect prognosis, including statins, other anti-platelet agents (mainly P2Y12 inhibitors) and coronary interventions.
Recent clinical trials have shown that aspirin does not show clinical benefit when given for pri-mary prevention. The duration of dual anti-platelet agent therapy after ACS and/or PCI is short-ened by the current guidelines, based on the results of recent trials. It also seems that in patients with stable coronary artery disease or those after PCI, long-term monotherapy with P2Y12 inhib-itors are better and safer than aspirin monotherapy. The most recent practice guidelines have changed the recommendations concerning aspirin.
Yet, aspirin loading for patients with ACS (including STEMI and NSTE-ACS) is still considered a “gold-standard”. Data from animal models suggest that aspirin loading can block myocardial protection by various agents. However, as discussed in this manuscript, the original evidence sup-porting aspirin loading is weak and has never been properly tested. It might be that aspirin load-ing can be skipped. Further clinical studies are needed to assess clinical outcomes in these patients without and with aspirin loading.”

Round 2
Reviewer 1 Report
Comments and Suggestions for Authors
I appreciated the chapter on cangrelor.
I still recommend to create two different chapters on DAPT duration (long/short) and secondary prevention. In the the present form is not clear.
Author Response
We modified the text. See attached file

Reviewer 3 Report
Comments and Suggestions for Authors
The authors responded satisfactorily to my requests and doubts, congratulations
Author Response
Thank you very much for the detailed review